# Recent Advances in Applications of Co-B Catalysts in NaBH₄-Based Portable Hydrogen Generators

**Valentina I. Simagina, Anna M. Ozerova** [ID]**, Oksana V. Komova** *[ID] **and Olga V. Netskina** [ID]

Boreskov Institute of Catalysis SB RAS, Lavrentieva Av. 5, 630090 Novosibirsk, Russia;
simagina@catalysis.ru (V.I.S.); ozerova@catalysis.ru (A.M.O.); netskina@catalysis.ru (O.V.N.)
* Correspondence: komova@catalysis.ru; Tel.: +7-383-330-7458

**Abstract:** This review highlights the opportunities of catalytic hydrolysis of NaBH₄ with the use of inexpensive and active Co-B catalysts among the other systems of hydrogen storage and generation based on water reactive materials. This process is important for the creation of H₂ generators required for the operation of portable compact power devices based on low-temperature proton exchange membrane fuel cells (LT PEM FC). Special attention is paid to the influence of the reaction medium on the formation of active state of Co-B catalysts and the problem of their deactivation in NaBH₄ solution stabilized by alkali. The novelty of this review consists in the discussion of basic designs of hydrogen generators based on NaBH₄ hydrolysis using cobalt catalysts and the challenges of their integration with LT PEM FC. The potential of using batch reactors in which there is no need to use aggressive alkaline NaBH₄ solutions is discussed. Solid-phase compositions or pellets based on NaBH₄ and cobalt-containing catalytic additives are proposed, the hydrogen generation from which starts immediately after the addition of water. The review made it possible to formulate the most acute problems, which require new sci-tech solutions.

**Keywords:** NaBH₄; H₂ generation; acid; Co-B catalyst; deactivation; solid-phase composition; pellet; reactor

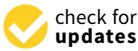



## 1. Introduction

The accumulation of electric energy in the hydrogen cycle is one of the main scenarios for the transition of the world power industry to carbon-free energy carriers. Electrochemical energy converters represented by fuel cells with a high efficiency (up to 70%) are proposed for the efficient involvement of hydrogen in the process of electric energy generation. Many efforts have been made for the development of power devices based on low-temperature solid-polymer fuel cells with proton exchange membrane (LT PEM FC). The rapid oxidation of H₂ and the formation of water as a sole by-product are advantageous for the creation of small-sized and low-weight devices [1]. However, there is a problem that hinders the commercialization of fuel cells in the power source market, namely, the lack of suitable methods for compact, safe, and efficient hydrogen storage [2–8]. This problem is most acute in the development of small-sized energy devices, which require compact gas generators providing a high yield of hydrogen per unit weight or volume without additional heating and LT PEM FC poisoning. The indicated requirements restrain the choice of hydrogen-generating materials as the potential energy carriers.

Conventional methods of hydrogen storage in compressed (15–70 MPa) or liquefied (−253 °C and 0.2 MPa) form are unacceptable for use in compact power sources due to the low hydrogen content in a unit volume (0.042 g·cm⁻³ at 70 MPa for compressed H₂ and 0.07 g·cm⁻³ for liquefied H₂), large sizes of cryogenic plants, high energy consumption, explosion hazard, and evaporation losses up to 3 vol% per day. It is known that hydrogen readily diffuses through metals, thus destroying them [9].

Among various hydrogen-containing compounds, a high gravimetric and volumetric content of hydrogen is typical of hydrides, particularly borohydrides [10–13]. First, they ex-

ceed other compounds in the content of hydrogen. For example, hydrogen density in $KBH_4$ is 0.083 $g \cdot cm^{-3}$; in $NaBH_4$, 0.112 $g \cdot cm^{-3}$; and in $LiBH_4$, 0.123 $g\ cm^{-3}$; these values are higher than the density of liquid hydrogen. Second, in the case of their interaction with water, the yield of hydrogen increases due to the involvement of water in the chemical process. However, the hydrolysis of hydrides is accompanied by the release of a great amount of heat per mole of the produced hydrogen [14,15]. The hydrolysis of $NaBH_4$ is characterized by a low thermal effect (53–80 kJ per mole of $H_2$ [15,16]); hence, the entire process is safe and controllable. This hydride has an acceptable price, and its hydrolysis products are environmentally benign [17,18]. In this connection, $NaBH_4$ has been thoroughly studied in the last 20 years as a compact form of hydrogen storage. The choice of acid accelerators and catalysts for the hydrolysis of $NaBH_4$ and the organization of the stable and controllable generation of $H_2$ in this process are the widely discussed subjects. A model where hydrogen is produced through the electrolysis of water, taking advantage of the electrical energy produced by a renewable generator (photovoltaic panels), and chemically stored by the synthesis of $NaBH_4$, is discussed in [19]. It was shown that due to the safety and high volumetric density of $NaBH_4$ it will be a promising technology for transportation to remote sites where hydrogen is released from $NaBH_4$ hydrolysis and used for energy production.

This review considers the main classes of water-reactive materials for hydrogen production. The focus is on hydrogen production via the $NaBH_4$ hydrolysis. Achievements in acid hydrolysis and catalytic hydrolysis with inexpensive and active cobalt catalysts are discussed, with the main emphasis on their transformations in the reaction medium. The new data on the preparation and investigation of solid-phase and pelletized compositions containing $NaBH_4$ and a cobalt catalyst are reported. The design of hydrogen generators based on $NaBH_4$ hydrolysis with cobalt catalysts and the problems of their integration with LT PEM FC are reviewed. The data obtained are generalized to outline main directions of further studies that are promising for the development of autonomous compact power devices based on LT PEM FC.

## 2. $NaBH_4$ as Water-Reactive Material for Hydrogen Storage and Production

The analysis of literature has revealed two main directions in the development of portable systems for hydrogen storage and generation: the physical storage of adsorbed hydrogen and the chemical storage of bound hydrogen. The chemical storage of hydrogen as stable chemical compounds is commonly characterized by high hydrogen density, minimum requirements to auxiliary infrastructure, and low energy consumption [20]. However, there are some problems: the sensitivity of materials to storage conditions, stability in iterative hydrogenation-dehydrogenation cycles, the regeneration of dehydrogenation products, the cost and accessibility of components of a hydrogen generation system, etc.

Current studies in the field are focused also on achieving the target values of volumetric (VHSC, $g \cdot cm^{-3}$) and gravimetric (GHSC, wt.%) hydrogen storage capacity, complying with the requirements to temperature and hydrogen release rate, and establishing a mechanism of the hydrogenation–dehydrogenation reactions, particularly the catalyst-assisted reactions. The materials under consideration can be classified as reversible or irreversible storage systems. In reversible systems, a material can be recharged with gaseous hydrogen directly "on board" [20–29], whereas irreversible systems are regenerated chemically, "off board".

Table 1 lists the main hydrogen storage systems based on hydrolysis or the involvement of water in the hydrogen generation process, in which the spent fuels or byproducts should be removed from the reactor and regenerated "off board". All these processes are exothermic; they often do not require an additional heat supply and can be performed at ambient temperature, which is an essential advantage. Here, it is difficult to perform the reaction with a high yield using a limited amount of water to provide high GHSC values [30]. Serious problems emerge with heat and mass transfer, which are related to intense heating of the reaction medium, the encapsulation of unreacted hydride with hydrolysis products, and the deposition of hydrolysis products on the reactor walls and catalyst surface, leading to catalyst deactivation. This is why the literature provides data mostly for the diluted solutions with low GHSC.

**Table 1.** Water-reactive materials for hydrogen storage.

| Material | Reaction | $\Delta H$, kJ/mol $H_2$ | Calc. GHSC, wt.% | Advantages | Disadvantages |
|---|---|---|---|---|---|
| Al [31] | $2Al + 6H_2O \rightarrow 2Al(OH)_3 + 3H_2$ | $-280$ (at ~50–100 °C) | 3.7 | + low cost, abundance on the earth<br>+ light weight<br>+ environmental-friendly products, may be raw materials of ceramics<br>+ improved and tunable properties at additions of dopants (salts, metals, oxides, hydrides, carbon materials, etc.), alloying with metals, reducing the particle size (BM), etc. | − surface passivation of native oxide layer (more relevant to Al)<br>− the hydrolysis products deposition on metal particles surface (more relevant to Mg)<br>− heat and mass transfer problem<br>− unpromising regeneration process of solid hydrolysis products |
| Mg [32] | $Mg + 2H_2O \rightarrow Mg(OH)_2 + H_2$ | $-354$ | 3.3 | | |
| $MgH_2$ [33] | $MgH_2 + 2H_2O \rightarrow Mg(OH)_2 + 2H_2$ | $-160$ | 6.4 | + low cost<br>+ can be industrially produced with a high energy efficiency<br>+ improved and tunable properties at addition of Brønsted acids, salts, metal catalysts, and alloying with rare earth elements, BM, etc. | − unstable in the presence of moisture<br>− reacts with water very slowly and incompletely<br>− formation of dense passivation layers of $Mg(OH)_2$<br>− complicated regeneration process $Mg(OH)_2$ to Mg (by reactive hydrogen plasma process) |
| NaSi [34] | $2NaSi + 5H_2O \rightarrow Na_2Si_2O_5 + 5H_2$ | $-35$ | 5.2 | + commercial availability<br>+ no activation procedure is required<br>+ by-product is of market value | − unstable in the presence of moisture<br>− use of liquid hydrocarbons for stabilization<br>− excess water is required<br>− problem with design of the reactor to control the rate of gas evolution |
| $NaBH_4$ [17,18,35] | $NaBH_4 + 2H_2O \rightarrow NaBO_2 + 4H_2$ | $-60$ | 10.8 | + commercial availability<br>+ high GHSC<br>+ safe and controllable on-board $H_2$ generation, commercialized generators<br>+ producing pure $H_2$ at ambient temperatures without energy supply<br>+ inflammable<br>+ high solubility (14.5 mol·L$^{-1}$): 30 wt.% $NaBH_4$ solution contains 6.7 wt.% $H_2$<br>+ improved and tunable properties at addition of catalysts or acids<br>+ concentrated water solutions or solid pellets with catalysts may be used as $H_2$ sources | − instability of aqueous solutions, stabilization with NaOH is required<br>− deactivation of catalysts<br>− hydrolysis products deposition on the catalyst surface and reactor walls in limited water conditions<br>− off-board and high-cost regeneration of solid hydrolysis products<br>− heat and mass transport problem at design of reactors for $NaBH_4$ interaction with stoichiometric amount of water |
| $KBH_4$ [36] | $KBH_4 + 2H_2O \rightarrow KBO_2 + 4H_2$ | $-55$ | 8.9 | + improved properties at addition of catalysts | − instability of aqueous solutions<br>− low solubility (3.5 mol·L$^{-1}$): low GHSC |

**Table 1.** *Cont.*

| Material | Reaction | $\Delta H$, kJ/mol $H_2$ | Calc. GHSC, wt.% | Advantages | Disadvantages |
|---|---|---|---|---|---|
| $LiBH_4$ [37] | $LiBH_4 + 2H_2O \rightarrow LiBO_2 + 4H_2$ | $-90$ | 13.8 | + high GHSC<br>+ improved properties at addition of catalysts, dopants (MWCNTs [1], $NH_3BH_3$), use of double-solvents, etc. | $-$ preparation from $NaBH_4$<br>$-$ flammable and unstable in the presence of moisture, instability of water solutions<br>$-$ high exothermic hydrolysis process<br>$-$ incomplete hydrolysis<br>$-$ hydrolysis products deposition on the catalyst surface and reactor walls in limited water conditions |
| $NH_3BH_3$ [29] | $NH_3BH_3 + 2H_2O \rightarrow NH_4BO_2 + 3H_2$ | $<-50$ | 9 | + high stability of aqueous solution<br>+ high solubility ($11.4 \text{ mol·L}^{-1}$)<br>+ improved and tunable properties at addition of catalysts | $-$ high cost<br>$-$ the presence of $NH_3$ impurity in the $H_2$ stream<br>$-$ deactivation of catalysts<br>$-$ hydrolysis products deposition on the catalyst surface and reactor walls in limited water conditions<br>$-$ off-board and high-cost regeneration of solid hydrolysis products |
| $N_2H_4BH_3$ [38] | $N_2H_4BH_3 + 3H_2O \rightarrow N_2H_4 + B(OH)_3 + 3H_2$<br>$N_2H_4BH_3 + 3H_2O \rightarrow N_2 + B(OH)_3 + 5H_2$ | 81.5<br>39 | 6<br>10 | + possibility of catalytic dehydrogenation of the $N_2H_4$ moiety<br>+ high theoretical GHSC<br>+ stability of water solutions at pH $\geq$ 7/5<br>+ improved properties at addition of catalysts | $-$ high cost<br>$-$ search for effective, low-cost, and stable catalysts for the hydrolytic dehydrogenation of both the $BH_3$ moiety and the $N_2H_4$ moiety<br>$-$ low solubility ($1.3 \text{ mol·L}^{-1}$)<br>$-$ lack of knowledge of the characteristics of the process in limited water conditions, providing GHSC |

The analysis of data listed in Table 1 revealed that sodium borohydride ($NaBH_4$) is the most promising inexpensive material for hydrogen storage in terms of the operating temperature, stability, and safety in air. In addition, its catalytic hydrolysis has been thoroughly studied with respect to hydrogen generation and the regeneration of hydrolysis products (Figure 1). Note that the involvement of water increases twofold the hydrogen yield upon hydrolysis as compared to thermolysis. Compact hydrogen generators based on sodium borohydride are being developed in many countries: USA [39–42], Japan [43], South Korea [44–46], China [43], France [47], Portugal [48,49], Italy [50,51], Taiwan [52], Russia [53,54], and others. A review of $NaBH_4$-based generators with the use of Co-containing catalyst will be given in Section 7. It should be noted that a somewhat greater interest in the hydrolysis of ammonia borane (Figure 1) is caused by the novelty of this compound and its high stability in air [55]; nevertheless, the high cost of ammonia borane and the presence of ammonia admixture in the evolved hydrogen limit its practical application. Thus, the review further considers studies on the systems for hydrogen generation and storage that are based on the hydrolysis of $NaBH_4$.

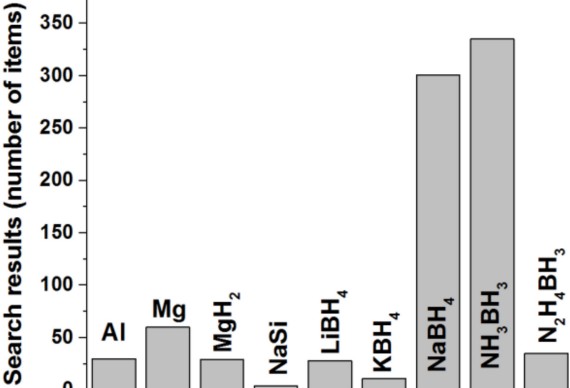

**Figure 1.** Number of results for a bibliometric resource search made on the Web of Science website (December 2020). The search combination was "compound" AND "hydrogen storage material" AND "hydrolysis". An exception was made for sodium silicide. In this case, the search combination was "compound" AND "hydrogen storage".

Sodium borohydride is known to be highly stable in the solid state in the absence of moisture and carbon dioxide [18]. Upon dissolution, it slowly reacts with water, releasing 4 moles of hydrogen per 1 mole of $NaBH_4$:

$$NaBH_4 + 4H_2O \rightarrow Na^+ + B(OH)_4^- + 4H_2\uparrow. \tag{1}$$

According to [16,39,56], a thermal effect of reaction (1) is 210–300 kJ/mol $NaBH_4$. In the case of insufficient heat removal and high concentration of $NaBH_4$, the temperature may increase, which may lead to a vigorous spontaneous release of $H_2$. The reaction product is sodium borate, the form of which depends on the pH of the solution: at pH < 9, $B(OH)_3$ prevails, whereas at pH > 9, the main species is $B(OH)_4^-$ [57]. If the reaction product is dried at a temperature below 110 °C, it is present as $NaBO_2 \cdot 2H_2O$ and $NaBO_2 \cdot 4H_2O$ [58].

The rate of hydrogen evolution in the spontaneous hydrolysis of sodium borohydride decreases with time, and the complete conversion of the hydride is not achieved. This is related to an increase in the pH of the reaction medium due to the accumulation of sodium borate. This property formed a basis for the stabilization of $NaBH_4$ aqueous solutions by an alkali (NaOH) [59,60], thus providing their long-term storage without considerable losses in the hydrogen capacity. A complete and rapid release of hydrogen from $NaBH_4$ solutions (both aqueous and stabilized by NaOH) is possible only in the presence of acids and catalysts.

### 3. Acid Hydrolysis of NaBH$_4$

This method was considered by Schlesinger et al. in the well-known paper published in 1953, which was devoted to the hydrolysis of sodium borohydride [61]. The development of hydrogen economy gave an impetus to investigation of such systems. Since the GHSC is among the key parameters [62], the most popular are such designs of hydrogen generators where a limited volume of acid solution with the optimized concentration is fed to solid NaBH$_4$ at a controllable rate [63,64]. The problems to be solved are the necessity of increasing the hydrogen yield under water deficit conditions and the formation of solid products hindering the contact between hydride and acid accelerator [65].

The application of strong inorganic acids (HCl, H$_2$SO$_4$) instead of weaker organic acids ensures a higher yield of hydrogen [63,65] and makes it possible to decrease the amounts of acid and water, thus providing high GHSC values. According to [66], the acid hydrolysis of NaBH$_4$ in the case of hydrochloric acid can be represented as follows (2):

$$NaBH_{4(s)} + xHCl + (x + y + 2)H_2O \rightarrow xNaCl + xH_3BO_3 + (1 - x)NaBO_2 + yH_2O + 4H_2. \tag{2}$$

The use of 2.74 M HCl at x = 0.19 and y = 4.74 gave GHSC = 5.1%, taking into account the weight of all the chemical reagents in the system. The optimization of conditions of this process was described in detail in [67].

A high aggressiveness of strong acids imposes substantial limitations on the materials used in the construction of hydrogen generators, so the application potential of weaker acids in this process is still being investigated. Thus, the design of a hydrogen generator proposed in [68] implies the interaction of a stoichiometric amount of water with a mixture of solid reagents NaBH$_4$ + H$_3$BO$_3$, which provides GHSC = 3.88%. Among the tested solutions of organic acids, high activity was observed for 12 N formic acid [65], acetic acid at a molar ratio CH$_3$COOH/NaBH$_4$ = 2 [63], as well as formic and malic acids at the concentration above 10% [64].

As shown in the literature, the process of acid hydrolysis of NaBH$_4$ can be arranged in different ways. For example, heterogeneous acid catalysts can be employed [69,70], thermal interaction of NaBH$_4$ with H$_2$C$_2$O$_4$·2H$_2$O is investigated [71], and the hydrolysis is performed using the ZnCl$_2$ + NaBH$_4$ composition, where hydrochloric acid is released in the reaction medium upon the hydrolysis of ZnCl$_2$ [72]. A combination of low-concentrated acids (0.01, 0.1, or 0.25 M H$_3$PO$_4$; 0.25 M HCl; 0.25, 0.1 M CH$_3$COOH) with catalysts (Ni, Cu) is used to hydrolyze the NaBH$_4$ solution stabilized by alkali [73–75].

The purity of evolved hydrogen is very important for the acid hydrolysis of NaBH$_4$. Conventional designs of the reactors include gas–liquid separators, water adsorbent, and acid vapor adsorbents [66,76]. Particular attention is paid to purification of hydrogen from alkali impurities that are carried by water in the gas flow [64]; however, a possible release of diborane (B$_2$H$_6$) is not mentioned. On the other hand, B$_2$H$_6$ can form via the interaction of sodium borohydride with inorganic acids [77]:

$$2NaBH_4 + 2HCl \rightarrow 2NaCl + B_2H_6 + 2H_2. \tag{3}$$

Certainly, in diluted solutions, diborane is instantaneously hydrolyzed with a release of hydrogen. Nevertheless, when acid hydrolysis is carried out in the highly concentrated solutions of sulfuric and hydrochloric acids, not only acid impurities but also diborane are released to the hydrogen-containing gas [78]. This problem of the acid hydrolysis of NaBH$_4$ deserves special attention.

### 4. Catalytic Hydrolysis of NaBH$_4$

The most promising method used to obtain high-purity hydrogen from NaBH$_4$ is its catalytic hydrolysis (1). Meanwhile, sodium borohydride is widely used as a reductant for obtaining highly dispersed metals and metalloids in a solution at ambient temperature [79], so the distinctive feature of catalysts for the hydrolysis of sodium borohydride is the possibility to form the active component from precursor compounds directly in the reaction medium under the action of the hydride (in situ).

Optimization of the catalyst makes it possible to control the hydrogen generation process [13,17,80–82] as well as to stop and launch it at the customer's request [39]. In addition, the application of catalysts allows hydrogen generation to be performed in a wide temperature range from −40 to +60 °C. The possibility of generating hydrogen at negative temperatures is provided both by the low freezing point of $NaBH_4$ solutions stabilized with an alkali [83] and by the heat that is rapidly released during the catalytic hydrolysis [16]. It should be noted that the produced hydrogen-containing gas does not contain diborane impurities, which were observed in the acid hydrolysis of $NaBH_4$ [78].

The catalytic hydrolysis of $NaBH_4$ is an intricate process, which is affected by the properties and concentration of catalyst, the concentration of $NaBH_4$, the addition of NaOH, and the reaction temperature. At this stage of investigation, the mechanism and kinetics of this process remain debatable [82,84–86]. Way back in 1971, K. A. Holbrook and P. J. Twist studied the hydrolysis of an alkaline solution of $NaBH_4$ by deuterated water in the presence of cobalt and nickel borides [87]. Mass-spectroscopic analysis of the evolved gas revealed that it contains one hydrogen atom from $NaBH_4$ and one deuterium atom from $D_2O$. The authors proposed the mechanism including dissociative chemisorption of the borohydride anion on the catalyst surface, charge transfer through the catalyst, and electrochemical decomposition of water as the key steps. This mechanism was verified by NMR spectroscopy for the Pd/C catalyst [88] and is used by researchers for interpreting various regularities that are observed in the hydrolysis of $NaBH_4$ [86,89–91]. The kinetics of the $NaBH_4$ catalytic hydrolysis is described using the Langmuir–Hinshelwood and Michaelis–Menten models, which take into account the step of $BH_4^-$ adsorption on the catalyst surface and the effect of temperature [84,85,92].

The literature presents many studies on the catalytic hydrolysis of $NaBH_4$ in the presence of catalysts containing noble metals [39,81,82,93–95]. According to independently performed studies [40,96,97], the highest activity in the $NaBH_4$ hydrolysis was shown by rhodium, ruthenium, and platinum compounds. The catalytic activity decreases in the following order: Rh > Ru > Pt >> Pd [97]; this sequence is retained also when metals are deposited on different supports. The research studies on Rh catalysts are restricted by their high cost. Investigation of the catalytically active phase formed from compounds of ruthenium, rhodium, platinum, and palladium in $NaBH_4$ medium has shown that it consists of agglomerated nanosized particles with a metal crystal lattice [98–100]. A residual amount of boron was presented in Ru metal [101]. The comparison of activity of noble and non-noble catalysts was discussed in [81,82]. It can be seen that the activity of non-noble Co-B-based catalysts (15,000–35,000 $mL_{H2} \cdot min^{-1} \cdot g^{-1}_{cat}$) reaches the activity of Ru or Pt catalysts. So, the high cost of platinum metals strongly limits their wide application in the hydrogen generation systems.

The catalytic hydrolysis of $NaBH_4$ is considered as one of the main commercially accessible ways of hydrogen generation and storage for LT PEM FC. In this connection, particular attention is paid in the literature to a search for cost-effective methods of hydrogen regeneration from the hydrolysis products—sodium borates [102,103], for example, by means of thermochemical [104–109] or mechanical activation methods [103,110–112]. The latter ones allow the process to be performed at room temperature, thus enhancing its energy efficiency.

## 5. Co-B Catalysts for the $NaBH_4$ Hydrolysis

### 5.1. The Formation of Active Component in the Reaction Medium (In Situ)

In 1953, Schlesinger et al. demonstrated [61] that the introduction of soluble chlorides of transition metals in a sodium borohydride solution results in the formation of a black precipitate, which is catalytically active toward the $NaBH_4$ hydrolysis. The activity of chlorides in this case decreased in the series $CoCl_2 > NiCl_2 > FeCl_2 > CuCl_2 > MnCl_2$. Later works on the catalytic hydrolysis of $NaBH_4$ confirm that the highest gas release rate is observed in the presence of cobalt compounds, whereas nickel and iron compounds are less active [61,82,91]. In this connection, many works in the last 20 years were devoted to

the bulk and supported cobalt-containing systems whose active components were formed in an NaBH$_4$ medium [13,86,113–115]. In distinction to platinum group metals, cobalt compounds are reduced with the formation of amorphous particles of the catalyst [116–118]. It is noted that the activity of the in situ synthesized catalytic systems exceeds that of the preliminarily (ex situ) synthesized systems [119–122].

Sodium borohydride is the most optimal reductant for cobalt compounds. This follows from a comparison of data on the reduction of cobalt chloride and oxide in the presence of NaBH$_4$ or another boron-containing hydride—ammonia borane (NH$_3$BH$_3$), which is known to be a weaker reductant. It was shown [120,123] that the use of sodium borohydride in the reduction of cobalt chloride provides a higher reduction rate and results in the formation of a highly dispersed active state of the catalyst (Figure 2). A lower rate of cobalt chloride reduction in NH$_3$BH$_3$ medium is confirmed by the presence of an induction period on the hydrogen generation curve (Figure 2). According to [123], the average size of the particles formed from cobalt chloride under the action of sodium borohydride is ca. 30 nm. When cobalt chloride is reduced in a solution of ammonia borane, this leads to the formation of large particles with the diameter from 150 to 500 nm and the predominant size of ca. 250 nm. On the other hand, particles with the size < 10 nm and a low aggregation degree were obtained upon the reduction of CoCl$_2$ by a NaBH$_4$ solution in the presence of ammonia borane, the hydrolysis of which leads to the formation of NH$_4^+$ ions that are likely to have a stabilizing effect [124]. It should be noted that the literature provides a thorough description of the effect exerted by various factors determining the characteristics of amorphous cobalt catalysts, which are formed in a sodium borohydride medium: cobalt salt precursors [91,114,119,122,125], pH of the medium [91,126,127], solvent [91,128], NaBH$_4$/Co ratio in solution [91,125,129], the order and manner of reagents introduction [127,129,130], changes in the composition of NaBH$_4$ + NaOH solution upon storage [131], etc.

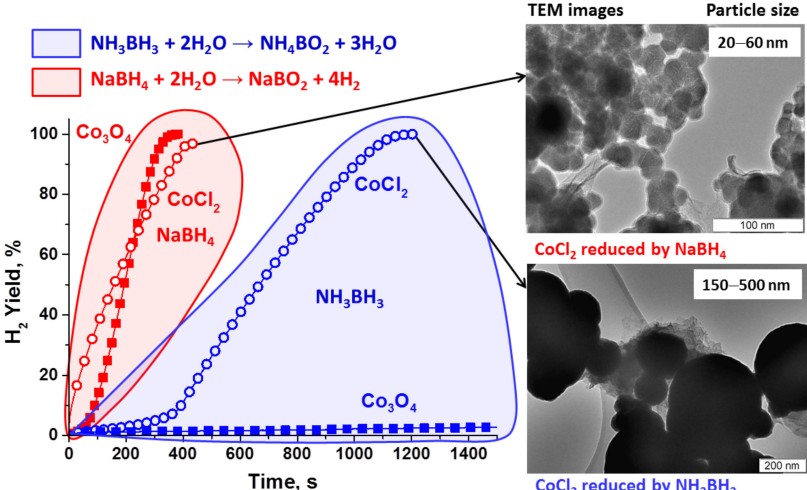

**Figure 2.** The influence of the nature of the hydride (NaBH$_4$, NH$_3$BH$_3$) on the formation rate and dispersion of Co-B catalysts forming in situ. The Co-B catalyst instantaneously formed in the NaBH$_4$ hydrolysis medium has a smaller particle size and shows a higher hydrogen generation rate. Conditions—1.2 mmol NaBH$_4$, 10 mL H$_2$O, 0.0117 g CoCl$_2$·6H$_2$O or Co$_3$O$_4$, 40 °C. Adapted from Refs. [123,132].

In distinction to salts, in the case of introduction of Co$_3$O$_4$ in both the NaBH$_4$ and NH$_3$BH$_3$ media, the induction period with a very low rate of gas generation is observed on the hydrogen generation curve (Figure 2). However, in sodium borohydride solution after a few minutes, H$_2$ generation starts and proceeds quite rapidly, whereas the rate of hydrogen evolution from ammonia borane solution remains insignificant even 6 h after the introduction of cobalt oxide [132]. It was found that within a short induction period,

$Co_3O_4$ is reduced under the action of $NaBH_4$ to form a new catalytically active ferromagnetic phase containing cobalt and boron [132]. Its accumulation in the course of reaction provided the growth of hydrogen generation rate. Thus, a correlation is observed between the activity and magnetic susceptibility of the cobalt catalyst that is formed in the reaction medium. Characteristics of the cobalt oxide samples (method of synthesis, morphology, defectness, crystallinity, etc.) have a substantial effect on this process [53,114,133]. For example, it was shown in [53] that defectness of the $Co_3O_4$ sample, which was synthesized by the calcination of cobalt chloride, provided its rapid reduction in the reaction medium, leading to the formation of the most active state of the catalyst. Significant differences were observed in the temperature dependences of magnetic susceptibility for such catalyst and the catalyst synthesized from the well-crystallized $Co_3O_4$. In addition, the presence of alkali in the reaction medium as well as sodium borate, which is accumulated due to the spontaneous hydrolysis of $NaBH_4$ upon storage of the solution, exerts a detrimental effect on the formation of the active phase from $Co_3O_4$ [53,134].

*5.2. The State of the In Situ Formed Active Component of Co-B Catalysts*

The nature of the catalyst active phase formed in the reaction medium from cobalt compounds during the hydrolysis of sodium borohydride is still a debatable question. Investigation of the active phase is complicated by the fact that catalysts are examined after their withdrawal from the reducing reaction medium followed by washing and drying. The number of publications on the subject is ever growing, and review papers summarizing new information are sometimes published [79,86,91,113,115,135].

It was shown that the catalyst particles formed in a sodium borohydride solution from cobalt compounds have a spherical shape [136,137] and are usually in the aggregated state [127,138,139]. The average size of the particles and their size distribution depend on the reduction conditions. For example, in [140], the synthesized samples had the particle diameter from 10 to 100 nm and a narrow particle size distribution with the maximum near 30 nm. Such particle sizes are most typical of the systems under consideration [127,137,138]. It is known that solutions of reverse micelles [136,141] or surfactants [142] used in the synthesis can prevent the aggregation of the particles and reduce their size (4–10 nm) [136,141].

The active phase of the catalysts, which are formed upon the reduction of cobalt compounds in a $NaBH_4$ solution, is X-ray amorphous and ferromagnetic. Its X-ray diffraction (XRD) pattern usually contains a low-informative broad line at $2\theta = 45 \pm 5$ [125,127,129,140,143]. In high-resolution transmission electron microscopy (HR TEM) data, it shows up as the diffuse rings (halo), indicating the amorphous structure of the particles [128,144,145].

It is well known that such catalysts contain Co, B, and O. The Co:B ratio is most often close to 2 [123,129,138–140], but it may vary depending on the reduction conditions of samples [113,139,146].

$$2CoCl_2 + 4NaBH_4 + 9H_2O \rightarrow Co_2B + 4NaCl + 12.5H_2 + 3B(OH)_3 \qquad (4)$$

Different content of oxygen is determined mostly by the degree of inertness of the atmosphere in which drying of the sample was performed [79]. According to HR TEM with energy-dispersive X-ray (EDX) analysis [123,139,147–149], nanosized particles of the catalysts are surrounded by a 2–5 nm thick shell containing oxygen compounds of cobalt and boron. This is confirmed by X-ray photoelectron spectroscopy (XPS) and infrared (IR) spectroscopy data indicating that the shell consists mostly of amorphous cobalt borate [139,149,150]. The formation of the shell may be related to the oxidation of an active component in an alkaline solution in the presence of sodium metaborate [131,139].

Indeed, according to XPS data, cobalt is present in the samples in two states, which correspond to $Co^0$ and $Co^{2+}$ in cobalt oxides, hydroxide and borate [113,149,151]. The $Co^0/Co^{2+}$ ratio considerably increases after the removal of the upper oxidized layer with a thickness no smaller than 3 nm by argon bombardment [147]. In distinction to cobalt, such treatment of the sample has only a slight effect on the ratio of oxidized and non-oxidized boron species when moving from the shell to the core [147,152].

Much attention is still focused on the composition and structure of the X-ray amorphous cobalt-containing core. In 2011, Arzac et al. published a paper [149] where examination by various methods (HR TEM, scanning TEM, electron energy loss spectroscopy, XRD, and XPS) revealed that the core of the 20–40 nm particle covered with the $Co(BO_2)_2$ shell contains 1–3 nm microcrystallites of metallic Co with the hexagonal crystal structure that are stabilized in the amorphous single- or multiphase matrix consisting of $Co_xB$, $Co_xO_y$, $Co(BO_2)_2$, and $B_2O_3$. The amorphous matrix prevents the oxidation and growth of cobalt clusters. Such a microstructure was observed in [153] for the nickel–boride system.

It should be noted that important additional information on the state of the core in amorphous catalysts containing $Co^0$ and $B^0$ can be obtained by magnetic methods. The residual magnetization typical of amorphous Co-B samples is lower than that of metallic cobalt and crystalline cobalt borides that are formed in the high-temperature synthesis [117,154]. However, their Curie temperatures ($T_c$) are very close to each other. For example, for amorphous cobalt borides with the molar ratio Co:B = 2:1 and Co:B = 3:1, $T_c$ is equal to 155 and 452 °C, respectively [155]. These values are close to $T_c$ of crystalline cobalt borides $Co_2B$ (156 °C) and $Co_3B$ (474 °C) [156]. The magnetic moment per Co atom and the Curie temperature decrease with increasing the boron content in the cobalt environment [155,157,158]. As shown by calculations [158], such weakening of ferromagnetism of the amorphous samples may be caused by changes in the electronic structure owing to hybridization of the electronic states of cobalt (s-d) and boron (s-p). Along with the boron content, magnetic properties of amorphous cobalt borides are affected also by their oxidation state [139], size [136,154], agglomeration of particles [154], and their thermal treatment [117,154]. The informativeness of this method concerning the state of Co-B catalysts synthesized in the medium of boron-containing hydrides was demonstrated in our previous works [53,132,147,148,159].

It was known that hydrogen enters the composition of samples that were reduced by $NaBH_4$ in organic solvent; in this case, the formation of bridge M–H–B structures with terminal B–H bonds was supposed [79]. For example, the reduction of cobalt chloride in ethanol led to the formation of a sample with the stoichiometry $Co_2B_1H_{0.6}$, the desorption of hydrogen from which was observed upon heating in vacuum up to 100 °C and higher temperatures [160]. Treatment of the sample with water was also accompanied by a release of gaseous $H_2$ and formation of $H_3BO_3$, which decreased the content of hydrogen and boron [160]. On the other hand, according to [116], the presence of hydrogen was detected in the composition of samples obtained in an aqueous solution. In that work, a series of Co-B samples was synthesized with the hydrogen content varying from 0.2 to 0.8 wt.%; therewith, the hydrogen content decreased with increasing the boron content in the samples. Further studies confirmed the presence of hydrogen in the sample obtained upon the reduction of cobalt chloride by sodium borohydride in an aqueous solution [161]. The molar ratio of elements was Co:B:O:H = 3.2:1.5:1.4:1 (or 0.46 wt.% H). Hydrogen release on the differential thermal analysis curve corresponded to the maximum at 230 °C [161], which is the decomposition temperature of higher borohydrides [162].

The magnetic susceptibility technique, XPS, EXAFS, XRD, and thermal analysis [147,161] were used to refine the model of the X-ray amorphous core, the structural elements of which include cobalt tetramers with the shortest Co–Co distance of ca. 2.5 Å and the coordination number equal to 2.7. It was supposed for the first time that such tetramers are stabilized in the hollows of a borohydride matrix. Taking into account X-ray diffraction data on the regular arrangement of cobalt clusters in the amorphous matrix (the distance between adjacent clusters is 5.8 Å), it was hypothesized that cobalt tetramers that formed in the sodium borohydride solution are involved in the build-up of the borohydride structure from $BH_4^-$ anions. The calculations by Density Functional Theory confirmed that ferromagnetic tetramers can exist in the borohydride matrix isolated from the environment by the amorphous shell, which may be the product of their decomposition in an aqueous medium. We think that further studies are needed to detail the state of hydrogen in such compounds.

### 5.3. The Role of Boron in the Activity of Co-B Catalysts for the NaBH$_4$ Hydrolysis

In 2016 [163], direct evidence indicating that boron is essential for the activity of cobalt catalysts in the hydrolysis of sodium borohydride was obtained. It was found that the introduction of boron via magnetron sputtering of cobalt on a nickel foil provided a more rapid evolution of hydrogen per cobalt atom as compared to the cobalt catalyst synthesized by a similar method without boron. The greater the boron amount in the sample, the higher the catalyst activity. This effect was attributed to an increase in the dispersion of the metallic phase. The replacement of boron by carbon in the catalyst synthesized by the same method led to a twofold decrease in activity in spite of the similar textural characteristics of cobalt–boride and cobalt–carbon catalysts. Therewith, the activity of cobalt–carbon catalysts increased during five cycles of their testing in NaBH$_4$ hydrolysis [163]. A comparative XPS study of these catalysts before and after the reaction demonstrated the accumulation of boron in the samples. Thus, the formation of the Co-B phase on the surface of the cobalt–carbon catalyst may be considered as the main factor enhancing its activity toward the hydrolysis of sodium borohydride.

A relation between the activity of the catalysts and boron content is noted also in some other studies [114,120,125,164]. This effect may be related to changes in the electronic state of cobalt. Contacting with boron enriches cobalt atoms with electrons, thus protecting cobalt from oxidation and activating BH$_4^-$ anions on the catalyst surface [120,143]. Therewith, calculations testify to the existence of the optimal stoichiometry between cobalt and boron, which provides high activity of the catalyst [165]. The site comprising two cobalt atoms and one boron atom is the most active one; this may be caused by the electronic structure of such a cluster, which provides the most favorable conditions for hydrogen binding.

### 5.4. Deactivation of the Co-B Catalysts in the NaBH$_4$ Hydrolysis

Stability of the cobalt-boride catalysts in long-term tests is very important for their practical application. Table 2 lists changes in the activity of bulk and supported Co-B catalysts during their cyclic testing and long-term operation. One can see that the indicated catalytic systems are often deactivated under the conditions of NaBH$_4$ hydrolysis. The activity of these catalysts changes to different extents, which may be related to differences in their physicochemical properties and testing conditions.

**Table 2.** Changes in the catalytic activity of bulk and supported Co-B catalysts during their long-term testing in the hydrolysis of NaBH$_4$.

| Catalyst | Test Conditions | The Number of Cycles or Test Duration | % of Initial Activity | Ref. |
|---|---|---|---|---|
| Co-B obtained in situ from CoCl$_2$ | 2% NaBH$_4$, 4% NaOH, 30 °C<br>19% NaBH$_4$, 4% NaOH, 30 °C | 3<br>3 | 89<br>89 | [166] |
| Co-B obtained in situ from LiCoO$_2$ | 0.5% NaBH$_4$, 40 °C | 6<br>14 | 100<br>40 | [159] |
| Co-Co$_2$B obtained by reduction of CoCl$_2$ in a methanol solution of NH$_3$BH$_3$ | 0.76% NaBH$_4$, 0,1% NaOH, 30 °C | 12 | 100 | [167] |
| Honeycomb Co-B obtained by plasma treatment of Co(NH$_3$)$_6$$^{2+}$ + KBH$_4$ + triethanolamine solution | 2% NaBH$_4$, 7% NaOH, 25 °C | 30 h<br><br>then | 100<br><br>deactivation | [168] |
| Co-B obtained in situ from Co$_3$O$_4$ | 10% NaBH$_4$, 5% NaOH, 25 °C | 3 h<br>20 h | ≈70<br>≈70 | [133] |

**Table 2.** *Cont.*

| Catalyst | Test Conditions | The Number of Cycles or Test Duration | % of Initial Activity | Ref. |
|---|---|---|---|---|
| Co-B in hydrogel obtained by impregnation with $CoCl_2$ solution and reduced by $NaBH_4$ | 0.2% $NaBH_4$, 5% NaOH, 30 °C | 5 | 93 | [169] |
| C-B/attapulgite clay obtained by impregnation with $Co(NO_3)_2$ and reduced by $NaBH_4$ | 10% $NaBH_4$, 5% NaOH, 25 °C | 9 | 31 | [170] |
| Co/nickel foam obtained by electrodeposition from $CoCl_2$ | 10% $NaBH_4$, 2% NaOH, 80 °C | 2<br>3 | 53<br>35 | [171] |
| Co-B/Pd-nickel foam obtained by dip-coating from $CoSO_4$ and reduced by $NaBH_4$ | 20% $NaBH_4$, 0.4% NaOH, 30 °C | 70 | ≈100 | [172] |
| Co-B/nickel foam obtained by electrodeposition from $CoSO_4$ | 5% $NaBH_4$, 5% NaOH, 25 °C<br>10% $NaBH_4$, 5% NaOH, 25 °C | 4<br>5 h<br>60 h | ≈88<br>≈60–70<br>≈60–70 | [173] |

Researchers suppose that activity of the cobalt–boride catalysts may decrease due to agglomeration of the particles [159,174,175] caused by their magnetization [140]. In the case of supported catalysts, a decrease in the hydrogen release rate during the long-term testing may occur also due to leaching of the active component under vigorous release of $H_2$ [176,177].

One of the main reasons of deactivation of the Co-B catalysts is the formation on their surface of a layer of strongly adsorbed borate anions [175]. It is difficult to remove this borate layer from the catalyst surface by washing with water [171]. Only the treatment with a diluted solution of sulfuric acid made it possible to destroy the borate layer of deactivated Co/nickel foam catalyst and provide a constant rate of hydrogen evolution in the next five cycles [171]. A similar result was obtained in [178] with the use of diluted hydrochloric acid.

The authors of [168] managed to restore the initial activity of the honeycomb Co-B catalyst after its operation for 30 h by a repeated plasma treatment. This effect was attributed to the reduction of the oxidized boron residing on the surface:

$$B_2O_3 + 6e^- + 3H_2O \rightarrow 2B + 6OH^-. \tag{5}$$

It should be noted that a prolonged contact of Co-B catalysts with an aqueous medium generally exerts a detrimental effect on their activity. It is known from the early works [138,179] that in water, cobalt borides are hydrolyzed to form boric acid and may completely transform into metallic cobalt upon boiling:

$$Co_2B + 3H_2O \rightarrow 2Co + H_3BO_3 + 1.5H_2. \tag{6}$$

The stability of transition metal borides directly depends on the content of boron [131,173]. Leaching of boron from the Co-B catalyst decreases its activity [114,164]. It is known that metallic cobalt nanopowders are low active in the hydrolysis of $NaBH_4$ [119,146]. In addition, a higher magnetization typical of $Co^0$ will lead to a greater aggregation of the catalyst particles in the reaction medium, thus decreasing the total surface area of the catalyst contacting the reaction medium [114,165].

XRD data obtained in our works show that the formation of $Co^0$ and $Co(OH)_2$ occurs when Co-B catalysts are tested in the $NaBH_4$ solution stabilized with an alkali [114,131,180]. The application of IR spectroscopy allowed us to reveal the presence of a certain amount of other oxidized cobalt compounds ($Co_3O_4$, CoO, CoOOH) in the catalysts [114]. The surface

oxidation of cobalt boride to cobalt hydroxide (7), when the pH of the solution increases due to accumulation of $NaBO_2$, was also reported in [139,175]:

$$Co_2B + OH^- + 7H_2O \rightarrow 2Co(OH)_2 + B(OH)_4^- + 3.5H_2. \tag{7}$$

A contact with a fresh portion of $NaBH_4$ may lead to reactivation of the catalyst due to reduction of the oxidized cobalt compounds, as was supposed in [139]:

$$2Co(OH)_2 + BH_4^- \rightarrow Co_2B + 0.5H_2 + OH^- + 3H_2O. \tag{8}$$

However, actually, there are only a few reports on the formation of the catalyst active phase from cobalt hydroxide under the action of $NaBH_4$. Thus, the authors of [127] observed under certain conditions ($CoCl_2$:$NaBH_4$:$NaOH$ = 1:6.5:1.9 in moles) the formation of colloidal $Co(OH)_2$, the further reduction of which resulted in the formation of highly reactive ultradispersed Co-B.

The majority of studies show that cobalt hydroxides, both neat and with intercalated anions of precursor salts, are characterized by a slow reduction in the $NaBH_4$ medium and hence by a low activity [114,119,122,150]. Raising the pH of the $NaBH_4$ solution extends the induction period during which the active Co-B phase is formed from cobalt compounds in the reaction medium [53,131,134].

The oxidation of cobalt and accumulation of borate anions in the reaction medium are accompanied by the formation of cobalt borate, which, according to [129,130], can be summarily expressed as follows:

$$Co^{2+} + 2NaBO_2 \rightarrow 2Na^+ + Co(BO_2)_2. \tag{9}$$

However, in distinction to cobalt hydroxide, cobalt borate is reduced quite readily after the introduction of fresh $NaBH_4$ solution to form the active Co-B phase [150]. The decrease in the activity of Co-B catalysts upon treatment with water or sodium borate solution was not so pronounced as in the case of their treatment with an alkali [114]. This occurs because changes in the phase composition of the catalyst follow different routes. Leaching of boron from the catalyst under the action of water or sodium borate solution is weaker than in the case of alkali solution. In addition, the oxidized cobalt compounds (cobalt borate and oxide) are formed, which transform more readily into the active state upon the addition of a fresh portion of hydride [132,150].

## 6. Hydrogen Storage Systems Based on Solid-State $NaBH_4$ Composites with Cobalt Catalysts

As it was discussed above in Section 5.4, cobalt-boride catalysts rapidly deactivate at high pH during the hydrolysis of $NaBH_4$ solutions stabilized by alkali. One of the ways to provide sustainable hydrogen generation is to abandon the use of alkaline solutions. Some authors propose solid-phase hydrogen storage systems based on $NaBH_4$ in a composition with cobalt catalyst (or its precursor), which include the dosing of a limited amount of water that initiates hydrogen release [47,181,182]. Another variant implies the dosing of a cobalt chloride solution on solid $NaBH_4$ [43]. The obtained results listed in Table 3 indicate that optimization of the water amount, the content of cobalt catalysts, and the method of its introduction in the reaction medium allow high GHSCs to be achieved, although the 100% conversion of $NaBH_4$ was not reached.

The application of pellets containing $NaBH_4$ and anhydrous $CoCl_2$ for hydrogen generation with the addition of water was first proposed by Schlesinger et al. [61,183], who noted that a decrease in the amount of introduced water leads to undesirable strong foaming and heating of the reaction medium. This study was further developed within the advancement of hydrogen economy. The application of the pellets containing $NaBH_4$ and inexpensive cobalt catalyst has some advantages: a high content of hydrogen in a pellet (up to 10 wt.%), storage under moisture-proof conditions without hydrogen losses, the absence

of hydride dust during the operation, and convenient application because gas generation starts immediately after water addition. Water can be taken from any natural source.

**Table 3.** Characteristics of $H_2$ evolution from the solid-phase system based on $NaBH_4$.

| Solid Phase Composition | $H_2O:NaBH_4$ in Moles | Conversion (%) | GHSC (wt.%) | Reference |
|---|---|---|---|---|
| 80 wt.% $NaBH_4$, 20 wt.% $CoCl_2$ | 9:1 | 78 | 3.4 | [47] |
| 85 wt.% $NaBH_4$ 16 wt.% $CoCl_2 \cdot 6H_2O$ [1] | 4:1 | 81.2 | 6.7 | [43] |
| 87.5 wt.% $NaBH_4$, 12.5 wt.% $Co^{2+}$/IR-120 | 4:1 | 90.95 | 6.7 | [181] |
| 90 wt.% $NaBH_4$, 10 wt.% $Co_2B$ | 3:1 | - | 8.7 | [182] |

[1] a $CoCl_2$ solution was dosed.

It was found that the pressing of a solid mixture of the hydride and catalytic additive as well as the uniform distribution of components over the composition substantially increase the hydrogen evolution rate [47,180,181]. The introduction of a precursor ($CoCl_2 \cdot 6H_2O$) into sodium borohydride is more efficient in comparison with the pre-reduced Co-B catalysts because the formation of the active phase, which starts in the pressing step, provides a more active state of the catalyst [184,185]. It should be noted that the activity and dispersion of the catalyst formed under the action of $NaBH_4$ are strongly affected by the amount of coordinated water in the composition of chloride hydrates (Co, Ni), which are used to prepare the pellets [186]. The rate of hydrogen generation can be controlled by varying the amount of catalytic additive [53] and the dispersion of the Co-B catalyst [180,187,188].

Current studies also reveal a considerable heating of the reaction medium (>100 °C) in the catalytic hydrolysis of $NaBH_4$ performed with a small amount of water. This allows hydrogen to be generated at low ambient temperatures without additional heat input [61,182]. It is noted that the extent of heating of the reaction medium depends on the content of water [61,181] and catalyst [43,182]. As shown by the study reported in [43], a high $H_2$ yield at a low content of water can be reached only upon its heating up to the melting point of $NaBO_2 \cdot 2H_2O$ ($\approx$60 °C). On the other hand, heating of the reaction medium to the dehydration temperature of $NaBO_2 \cdot 2H_2O$ (>100 °C) increases the amount of water involved in the hydrolysis of $NaBH_4$. This was observed in [182], where a high GHSC equal to 8.7 wt.% was reached. This value exceeds the yield of hydrogen according to the stoichiometry of reaction (1) (see Section 2), which theoretically corresponds to GHSC = 7.3%. The authors attribute this to the formation of $NaBO_2 \cdot nH_2O$, where n < 2.

## 7. Hydrogen Generators Based on Hydrolysis of $NaBH_4$ Using Cobalt Catalysts

Fuel cells as electric energy generators are being applied now in unmanned air vehicles (UAV), warehouse equipment, submarines, householding, stationary battery backup, etc. They are simple in design, have zero atmospheric emission, high energy density, and high efficiency because chemical energy is converted directly to electric power. In addition, fuel cells are characterized by weak vibration, low noise, and low maintenance requirements [189]. Prototype power plants using LT PEM FC and hydrogen generator based on the hydrolysis of sodium borohydride in the presence of cobalt catalysts are reported in the literature (Table 4).

**Table 4.** The reported hydrogen generators with Co-containing catalysts.

| Hydrogen Generator Design | Catalyst | Conditions | Operating Time | Comments | Output Characteristics | Ref. |
|---|---|---|---|---|---|---|
| Semicontinuous reactor, $H_2$ washing flask (water) | Co-B/Ni foam | 19% $NaBH_4$, 4.5% NaOH, 2.5 mL/min | 3 cycles of 60 min | Stable $H_2$ generation with cat. re-activation between cycles | 1160 mL $H_2$/min, conversion 90%, start-up time [1] < 5 min | [190] |
| Flow catalytic reactor, gas–liquid separator, mesh filter, $H_2$ dehumidifier (silica) | Co/$Al_2O_3$ pellets | 15% $NaBH_4$, 5% NaOH, 3 mL/min | 2.5 h UAV flying test | Crushing $Al_2O_3$ | 946 mL $H_2$/min, integration with 100 W PEMFC stack, specific energy density 165 Wh/kg | [185,186] |
| Flow catalytic microreactor, hydrogen separator with gas–liquid separation membrane | Co-P-B/Ni foam | 10% $NaBH_4$, 5% NaOH, 0.06 mL/min | 10 cycles of 30 min | Decrease in max power output by 8% in 6 cycles | 16.1 mL $H_2$/min, conversion 98.8%, integration with micro PEMFC, max power output 174.6 mW | [191] |
| Flow catalytic reactor with cooling fans, volume-exchange fuel tank (fuel and spent fuel exchange the volume within a single fuel tank) | Co-B/ ISOLITE monolith | Alkaline 15% $NaBH_4$, 2.5 mL/min | 2 cycles of 30 min with 5 min rest in between, 1 h UAV flying test | Restart of $H_2$ generation with quick start-up time and decrease in rate by 9% | 1331 mL $H_2$/min, integration with 100-W PEMFC stack | [192] |
| All-in-one reactor (cat., chamber of pressurized $H_2$ (4–5 bar) and by-product separator are combined in single space) with cooling devices, $H_2$ wash tank; periodic pressure regulation be $NaBH_4$ supply; autonomous operation | Co-B/Ni foam | Alkaline 25% $NaBH_4$ | 2 cycles of 5 min | Increase in start-up time, stable pressure in $H_2$ chamber with cycles; cat. washing between cycles; stable HGR [2] and temperature during 17 min operation | ≈1200 mL $H_2$/min, integration with 100 W PEMFC stack, total energy density 360 Wh/kg | [193] |
| Flow catalytic reactor with cooling jacket, by-product level sensor, $H_2$ cooling device, resin for capture of sodium from $H_2$; removed water fed back to the reactor | Co-Fe-Ni powder | 20% $NaBH_4$, 3% NaOH | 20 cycles of 2 h, 3 h FC stack operating | Decrease in HGR by 1% in 10 cycles, constant rate over next 10 cycles | 5400 mL $H_2$/min, integration with 200 W PEMFC stack, total energy density 325 Wh/kg | [194] |
| Reactor with inner magnet attracting the magnetic cat. powder, H2 filter and drying device; recycling spent fuel back to fuel tank | Modified comm. Co catalyst | 20% NaBH4, 10% NaOH, 10 mL/min, preheated to 35 °C | 20 h | Good cat. stability | 25 mL $H_2$/min, conversion 90% | [195] |
| $\pi$-shaped catalytic reactor, gas–liquid separator, $H_2$ dehumidifier (silica) | $CoO_x$ + Ni + PVDF [3]/Ni foam | Alkaline 15% $NaBH_4$, 0.5–2 mL/min | | Improved characteristics compared to conventional reactor via rapid $H_2$ discharge from cat. surface | conversion 90.2% | [196] |

**Table 4.** *Cont.*

| Hydrogen Generator Design | Catalyst | Conditions | Operating Time | Comments | Output Characteristics | Ref. |
|---|---|---|---|---|---|---|
| Flow catalytic reactor, by-product separator, $H_2$ wash (water) and dehumidifier (silica) tanks | Co-P/Ni foam | 20% $NaBH_4$, 5% NaOH, 5.22 mL/min, preheated to 30 °C | 3 h | Cat. high durability | 2670 mL $H_2$/min, conversion 90%, start-up time 71 s, integration with 200 W PEMFC stack, energy density 252.1 Wh/kg | [44] |
| | | 20% $NaBH_4$, 1% NaOH, 2.5 mL/min for start-up, 12.9 mL/min next, thermostated to 20 °C | 2 cycles of 2 h with 2 h rest in between | Restart of $H_2$ generation with the same start-up time (2.5 min) and decrease in rate by 12.4%; $NaBO_2$ deposition on cat. | ≈6500 ml$H_2$/min, conversion 94%, start-up time 2.5 min | [197] |
| | | 15–18% $NaBH_4$, 5% NaOH, 3.07–3.5 mL/min, preheated to 30 °C | 2 cycles of 1 h, 3 h FC stack operating | Cat. washing and drying in air between cycles; increase in start-up time, decrease in conversion by 42.5% in 2nd cycle; stable $H_2$ generation during cycle | 1570 mL $H_2$/min, 97.8% conversion, integration with 100 W PEMFC stack, specific energy density 185.2 Wh/kg, max power 95.96 W | [198] |
| Flow catalytic reactor, liquid–gas separator with purge pump, $H_2$ cooling device (finned container with water + cooling fan), $H_2$ dehumidifier (silica) | Co/$Al_2O_3$ on the top of Co-P/Ni foam | 20% $NaBH_4$, 12 mL/min | 2 h | 2 catalysts provide fast start-up | 5900 mL $H_2$/min, integration with 500 W PEMFC stack, specific energy density 211 Wh/kg | [199] |
| Packed-bed catalytic reactor with two exit channels (top for $H_2$ and side for byproduct) placed in water bath (70 °C), 2 water traps, dessicator | Co-B/ IR-120 | 5% $NaBH_4$, 2% NaOH, 2.4 mL/min for start-up, 1.02 mL/min next | 3 cycles for ≈30 min | Modeling and validating a strategy of combining reactant feed rates for instant and stable $H_2$ generation | 130 mL $H_2$/min, conversion 99.3%, start-up time 70 s | [200] |
| Batch reactor with $NaBH_4$ pellets, injection of $Co(NO_3)_2$ solution, $H_2$ separator | Co-B reduced in situ | 1.6 g of $NaBH_4$ pellets, 2 mL of 0.1 M $Co(NO_3)_2$ solution | 30 min | Instable HGR | Max 913.5 mL $H_2$/min, 100% conversion, integration with 100 W PEMFC stack | [201] |
| Batch reactor with pills of solid-state $NaBH_4$ composites, rapid injection of water | Co-B/IR-120 reduced in situ | $NaBH_4$ (3 g), cat. (0.6 g) and silicon rubber (2.5 g) mixed in 20 solid-state pills composites; 20 g $H_2O$ | 2 h | Regulation HGR depending on the composition, shape and number of solid-state composites | 25 mL $H_2$/min, integration with 2 W PEMFC stack to power cellular phone | [52] |
| Batch reactor with cat., heat exchanger, $H_2$ buffer tank, batch control module | $CoO_x$/Ni foam | 15% $NaBH_4$ | 6 cycles for 6 min; 1 h FC stack operating | Efficiency depends on $NaBH_4$ concentration, volume of bath reactor and algorithm of next batch addition | Max 128 L $H_2$/min, conversion > 90%, integration with 3 kW PEMFC stack | [202] |

[1] Start-up time—the time for stabilization constant hydrogen flow. [2] HGR—hydrogen generation rate. [3] PVDF—polyvinylidene fluorid.

To provide the operation of PEM FC, pure hydrogen should be fed at a specified rate. This imposes certain requirements on the design of hydrogen generators and catalysts. As seen in Table 4, the designs ensuring a wide variation of hydrogen feed rate (from several milliliters to more than 100 L $H_2$ per minute) have already been proposed for fuel cell batteries of different power levels intended for remote, portable, and mobile systems—2 W [52], 100 W [192,193,198,201,203,204], 200 W [44,194], 500 W [199], 3 kW [202].

The published studies are devoted mostly to the development of hydrogen generators with a catalytic flow reactor, where an $NaBH_4$ solution stabilized by alkali is fed to the catalytic bed. The hydride solution is stored in a separate reservoir at room temperature and pumped into a reactor at a specified rate. In most cases, temperature is not controlled, and the hydrogen generation rate is determined by the contact time of $NaBH_4$ solution with the catalyst and by its characteristics. The most common scheme of the setup is displayed in Figure 3.

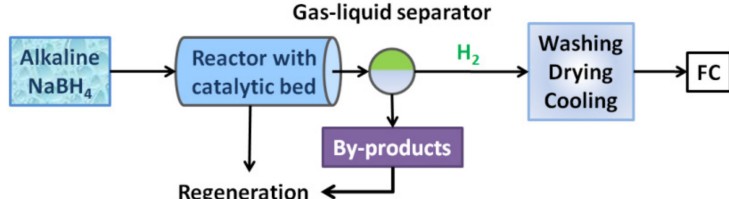

**Figure 3.** A schematic diagram of the hydrogen generation system with a flow reactor.

Such arrangement of the process is complicated by safety regulations for fuels of the irritant nature. A more serious problem related to the use of $NaBH_4$ solutions stabilized by alkali is the low GHSC value due to the great weight and dimensions of the fuel tank. The hydrolysis product—sodium metaborate—has a limited solubility in water, which decreases with an increase in pH [205]. To prevent its deposition on the catalyst and equipment, it is necessary to optimize the ratio of the hydride, water, and alkali in the solution [205]. This was made in [190], where a solution with the optimized composition (19% $NaBH_4$ + 4.5% NaOH) was fed to the Co-B/Ni foam catalyst in order to produce hydrogen for the functioning of 60 W-scale LT PEM FC. The GHSC value of 3.5 wt.% was reached (the calculation took into account weights of the hydride solution and catalyst), which is quite high for such systems. However, this value is much lower than the United States Department of Energy targets [62]. The authors of [197] observed also the deposition of sodium borate in the case of 20 wt.% $NaBH_4$ + 1 wt.% NaOH solution. The problem was solved by lowering the $NaBH_4$ concentration to 18 wt.% [198]. Even in the presence of alkali, the hydride slowly interacts with water [206], which is accompanied by a gradual loss in hydrogen capacity. For example, after storage of the 15% $NaBH_4$ + 5% NaOH solution for a year, the content of $NaBH_4$ decreased by a factor of 3, and a partial deposition of the formed $NaB(OH)_4$ (or $NaBO_2 \cdot 2H_2O$) was observed [131].

Another essential aspect is associated with the characteristics of the catalyst. First, when hydrogen generators are functioning in a flow-type ON/OFF mode, it is preferable to use macrostructured granular or monolithic catalysts in order to decrease the gas and hydrodynamic resistance in the system. In addition, such catalysts can easily be separated from the reaction medium and removed from the reactor for replacement. Thus, in [192], a Co-B catalyst supported on ISOLITE ceramic material shaped as the reactor volume was successfully used. In [200], cation-exchange polymeric resin Amberlite IR-120 was used as a support for the Co-B catalyst. The authors of [203,204] used a highly active granular $Co/\gamma$-$Al_2O_3$ catalyst; however, in the long-term tests, they observed poor durability and adhesion of the supported active component as well as the destruction of the support grains. The most popular support is Ni foam, which not only demonstrates high stability in aggressive reaction medium but also provides efficient heat transfer over the reactor volume [190,193,202]. To increase the adhesion between cobalt-boride coating

and Ni foam, the catalyst is heat-treated in an inert medium at moderate temperatures of 200–300 °C [190] and subjected to electroless plating [191,193].

A different approach to mounting a catalyst in the reaction zone was proposed in [50,195]. Magnets were placed in the reactor to hold a powdered ferromagnetic cobalt catalyst and prevent its washout with the liquid flow.

However, along with the washout of the catalyst active component, there is a problem of chemical deactivation of Co-B catalysts in the $NaBH_4$ alkaline solution, as was shown in Section 5.4. To solve the problem, in [190], each cycle of hydrogen release was followed by reactivation of the Co-B/Ni foam catalyst—it was washed with water and immersed in the $NaBH_4$ solution to reduce the oxidized cobalt compounds that formed upon testing. As a result, reusing of the catalyst under these conditions has been demonstrated at least three times without any deterioration of performance. A simple water washing of the catalyst removes the deposited borates from its surface, but it cannot restore the catalytic activity after a long-term cycle [198].

To extend the durability and enhance the activity, catalytic cobalt systems undergo a chemical modification. For example, Korean researchers produce hydrogen generators with the phosphorus-doped catalysts [44,191,197,198]. According to [207], the introduction of phosphorus increases the concentration of Co active sites on the catalyst surface. Another widely employed approach to improving the catalytic properties of cobalt catalysts is based on doping with transition metals. For example, the Co-Fe-Ni catalyst showed a superior stability during 20 hydrogen generation cycles of 2 h each [194].

In most cases, hydrogen generation is carried out without an additional heat supply because the hydrolysis of $NaBH_4$ is an exothermic process. Self-heating is observed in a catalytic reactor; its rate and temperature depend on the catalyst activity [44] and the hydride solution feed rate [197]. Under certain conditions, such self-heating can exceed 100 °C [202] and lead to the evaporation of water. This creates several problems. A deficit of water in the system will result in an incomplete release of hydrogen and a decrease in GHSC. In addition, due to an intense evaporation of water, undissolved sodium borates will actively deposit on the equipment and on the catalyst surface [208]; sodium borates can also be released as impurities into the gas phase together with water vapor [64], thus deteriorating the operation of the Nafion membrane in fuel cells. To lower the temperature, the reactor is equipped with cooling fans [192,193], and the released hydrogen is purified. To this end, hydrogen is passed through reservoirs with water for removing water-soluble impurities [44,190,193,197,198] and then with silica gel for desiccation [44,193,196–198,203,204]. In [194], a resin was used for the capture of sodium from $H_2$. Sometimes, $H_2$ cooling devices are also employed—for example, a finned container with water and cooling fan [199].

Note that the optimization of reactor material of the hydrogen generator is one of the key tasks. On the one hand, the reactor material should meet the energy device requirements of light weight, small size, and high power density. On the other hand, to avoid the self-heating of the reaction medium and to control the hydrogen generation rate, materials with high heat transfer properties should be applied. In addition, the reactor material should capable of withstanding pressure and hydrogen embrittlement.

It seems interesting that the contribution of a non-catalytic route of $NaBH_4$ hydrolysis will increase with the growth of reactor temperature. This suggests that the action of the catalyst is most important in the initial step, when the hydrogen generation process is launched. Indeed, the start-up problem is very essential. Thus, the target of the US Department of Energy's fuel cell start-up time is less than 5 s [62]. This means that the instantaneous hydrogen generation from the sodium borohydride hydrolysis reaction must be achieved. To minimize the time of establishing a stable generation of hydrogen, several approaches were suggested in the literature. The authors of [199] used two catalysts: $Co/\gamma\text{-}Al_2O_3$ was placed at the top of the reactor, while the remaining volume was filled with Co-P/Ni foam. Upon feeding, the $NaBH_4$ solution first contacted the highly active $Co/\gamma\text{-}Al_2O_3$, which increased the temperature and, accordingly, the rate of hydrogen evolution on the Co-P/Ni foam catalyst. In [197,200], the start-up time was reduced by

varying the feed rate of NaBH$_4$ solution. In the case of a highly concentrated 20 wt.% NaBH$_4$ solution, first, a low feed rate of the solution was used, and then, it was increased [197], whereas in the case of a 5 wt.% NaBH$_4$ solution, the process was initiated by the pulsed feeding of solution at a high rate, which was decreased later [200].

A design of the reactor also affects the hydrogen release parameters. The authors of [196] developed a π-shaped catalytic reactor, the shape of which facilitated a rapid removal of hydrogen from the catalyst surface, so as to increase the contact between sodium borohydride and the catalyst. This enhances the efficiency and stability of hydrogen generation in comparison with the reactor having a conventional shape.

Original configurations of hydrogen generators are proposed in order to decrease their weight and dimensions. In [192], the hydrogen generator featured a volume-exchange fuel tank, where the fuel and spent fuel exchange the volume within a single fuel tank. In [193], a novel concept using an all-in-one reactor was proposed in which a catalyst, hydrogen chamber, and by-product separator were combined in a volume. In addition, the reactor as well as a pump, cooling fans, valves, and controller were integrated in a single module. The produced hydrogen was not immediately sent to the fuel cell; it was stored in a hydrogen chamber at a pressure of 4–5 bar. The fuel injection frequency and hydrogen feed rate were controlled automatically in dependence on the electronic load. A sodium metaborate solution was discharged through a blow-down valve at a specified interval. Such a design considerably improved the characteristics of the response "fuel cell power—hydrogen feed rate—injection rate of new portions of the NaBH$_4$ solution", which provided a stable operation of LT PEM FC and simplified the utilization of sodium borate.

Note that all the flow reactors considered above make it necessary to use dispensers, which allow the control of the gas generation process but increase the weight and size of the hydrogen generators. A simpler instrumentation of the sodium borohydride hydrolysis process is typical of the batch (or autoclave) reactors (Figure 4). Their operation conditions allow catalysts to be used as powders instead of the macrostructured bed.

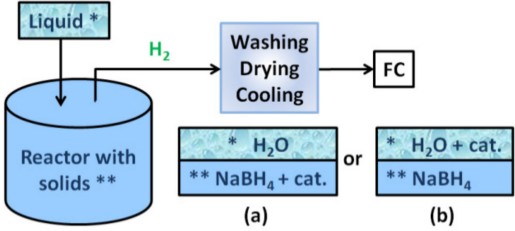

**Figure 4.** A schematic diagram of the hydrogen generation system with a batch reactor. Two approaches are possible: (**a**) injection of water into a reactor with solid-state mixture of NaBH$_4$ with a catalyst in the form of powder, pellets, or tablets; (**b**) injection of solution with catalysts or catalyst precursor into a reactor with solid NaBH$_4$; * indicates the composition of liquid phase; ** indicates the composition of solid phase.

In [201], hydrogen for the functioning of a fuel cell battery with a power of 100 W was generated in an autoclave reactor, where a cobalt nitrate solution was dosed by a syringe pump on NaBH$_4$ grains. However, this was accompanied by an unstable uncontrolled release of hydrogen with a pronounced hydrogen generation rate maximum, which was associated with a sharp growth of temperature in the reactor.

A constant rate of hydrogen generation without changes in the temperature at a rapid injection of water into spherical pills of a hydrogen-generating material was achieved in [52]. Along with sodium borohydride and a cobalt catalyst (Co$^{2+}$/IR-120), the composite included a hydrophobic silicone rubber, which facilitated a uniform penetration of water into the pills, thus providing a stable release of hydrogen. The only drawback is that the dilution of sodium borohydride with an inert polymer decreases the GHSC value.

The authors of [202] also used a batch reactor with the CoO$_x$/Ni foam catalyst to which a 15% NaBH$_4$ solution was fed. In addition to parametric studies, the authors developed

and successfully implemented a method for controlling time intervals between injections of fresh portions of the $NaBH_4$ solution into reactor in dependence on the PEM FC load. The developed hydrogen generator provided a continuous production of considerable amounts of hydrogen (123 L $H_2$/min) "on demand" to maintain the functioning of PEM FC with a power of 3 kW.

Thus, the main studies of hydrogen generators based on the catalytic hydrolysis of $NaBH_4$ are aimed to develop reactors with the advanced designs depending on the type of cobalt catalyst (supported, macrostructured, bulk, doped, or ferromagnetic catalysts or their precursors with subsequent reduction in the reaction medium). Extended parametric studies are usually performed for revealing the effect of catalyst properties and the method of its loading in the reactor unit, the flow rate, and the concentration of $NaBH_4$ solution as well as the time of its contact with the catalyst. This allows estimating the operational efficiency of a generator from different standpoints (HGR, conversion, start-up time, stability, durability, and integration with fuel cell). The results obtained made it possible to formulate the most acute problems, which require the development of the proposed sci-tech solutions and the creation of the new ones. Particularly, solutions are needed regarding the control of the temperature of reaction medium and its foaming in the reactor, stability of catalysts, purity of hydrogen under variable modes of generator functioning, weight and size characteristics of the generator, time of establishing a stable operation, and obtaining a rapid response to changes in the hydrogen demand of the fuel cell [30].

## 8. Conclusions

A significant increase in the number of publications in the last 20 years confirms that there is a great interest in the development of systems for hydrogen storage and generation based on sodium borohydride ($NaBH_4$). This is related to its commercial availability, price, safety, and the possibility to generate $H_2$ via hydrolysis at ambient temperatures. Studies on the development of such systems are carried out in several directions. Their main goal is a search for the methods that could increase the hydrogen capacity and control the gas generation rate. These characteristics, which are essential for the functioning of LT PEM FC, cannot be provided without the use of acids or catalysts.

The acid hydrolysis of $NaBH_4$ is the known process for which special reactors are designed; particular attention is paid to purification of the evolved hydrogen in a highly exothermic process. By now, the highest GHSC values have been achieved with the use of aqueous solutions of strong mineral acids (HCl and $H_2SO_4$). Their high aggressiveness imposes considerable limitations on the materials suitable for constructing hydrogen generators, so the possibility to use weaker acids in the process is being explored.

Judging from the number of publications, the catalytic hydrolysis of $NaBH_4$ should be considered as the process attracting the greatest attention. Catalysts based on noble metals (Rh, Ru, Pt) are known to exhibit the highest activity and stability; however, their high price initiates studies on the development of catalysts based on cobalt, which is the most active metal in the iron subgroup. Since $NaBH_4$ is a reductant, a considerable part of works is devoted to the investigation of nanosized amorphous cobalt–boride catalysts that are formed directly in the reaction medium of $NaBH_4$ hydrolysis from precursors (cobalt salts and oxides). Although there is quite a large body of experimental data concerning the composition and structure of the active components of such catalysts, it is still difficult to reliably describe the state of the active site of the Co-B catalyst, and further studies are performed to elucidate this issue.

The published results demonstrate that the application of Co-B catalytic systems in flow reactors for the hydrolysis of $NaBH_4$ alkaline solutions is limited by their strong degradation. So, a different method of hydrogen production in the $NaBH_4$ hydrolysis without an alkali was proposed for the Co-B catalysts. It implies the delivery of water to solid-phase or pelletized compositions of the hydride with a catalytic additive or feeding a solution of the Co-B catalyst precursor to solid $NaBH_4$. This process can be carried out in batch (or autoclave) reactors. Dosing a limited amount of water provides high

GHSC values as compared to the hydrolysis of $NaBH_4$ alkaline solutions in a flow reactor. However, in this case, there are problems of self-heating and foaming the reaction medium caused by the exothermicity of the reaction and vigorous gas formation. For estimation of the potential of such a hydrogen generation process, it is necessary to perform further experimental and theoretical studies on the effect of heat and mass transfer on the kinetics of hydrogen generation and its purity when optimizing the design of such generators and varying the conditions of their operating.

Thus, the main lines of investigation include the development and improvement of cobalt-based catalysts and reactors both for the generation of $H_2$ from $NaBH_4$ solution and for the case of solid-phase compositions based on $NaBH_4$. This will make it possible to optimize the weight and size characteristics of the reactor, as well as its heat exchange with the environment, to decrease the start-up time as well as the gas dynamic and hydraulic resistance in the catalyst bed, to solve the problems of catalyst washout and a loss in stability, to enhance the completeness of $NaBH_4$ conversion and increase the purity of evolved hydrogen, and, which is most important, to find flexible solutions for controlling the hydrogen generator capacity in dependence on the power of LT PEM FC and the consumer's goal. When creating the infrastructure for the application of high-purity hydrogen generation from $NaBH_4$, particular attention should be paid to the convenience and safety of the process (the prevention of flooding of the alkaline reaction medium, decreasing the surface temperature of hydrogen generator, collecting the products and catalysts for their regeneration, etc.) as well as to environmental and economic aspects of the proposed technologies.

**Author Contributions:** Conceptualization, V.I.S.; investigation, V.I.S., A.M.O. and O.V.K.; writing—original draft preparation, A.M.O. and O.V.K.; writing—review and editing, V.I.S. and O.V.N.; visualization, O.V.N. All authors have read and agreed to the published version of the manuscript.

**Funding:** This work was supported by the Ministry of Science and Higher Education of the Russian Federation within the governmental order for Boreskov Institute of Catalysis (project AAAA-A21-121011390006-0).

**Conflicts of Interest:** The authors declare no conflict of interest.

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
