# Peer review of "Recent Advances in Applications of Co-B Catalysts in NaBH4-Based Portable Hydrogen Generators"

_catalysts, doi:10.3390/catal11020268_

Round 1

Reviewer 1 Report

The present review article the catalytic hydrolysis processes of NaBH4with Co-B based catalysts for hydrogen storage and generation applications. The authors discuss further the potential for manufacturing of portable energy generator devices as well as the challenges in the integration with low temperature proton exchange membrane fuel cells. Overall, it is well referenced and structured and gives to the reader an overview of this research topic.  

I have a few points for the authors to consider.

  1. The manuscript has numerous grammatical and linguistic errors and should be checked and revised thoroughly. There are parts that the message is not only unclear but also highly confusing. Eg. Abstract: “…the place of catalytic hydrolysis of NaBH4with the using…”; “…the prospect of using alkali-free solid-phase compositions…”; etc. This is not an exhaustive list since similar errors occur throughout the manuscript.

  1. Page 2, line 88: “…examples of irreversible “off-board”…”. This part is confusing for the readers. What exactly ”off-board” means in this case? Please revise or provide additional explanation.

  1. Conclusions, line 729: “…heat and mass transfer and foaming on the kinetics of hydrogen…” This is slightly confusing. The foaming I would assume is due to exothermicity of the reaction and the parameters that might have some correlation with the kinetics of the H2 production rather than the actual foaming of the system. I: would suggest the authors revising this part.

The present review article discusses an interesting catalytic function with potential application in H2 generators and miniaturisation efforts for the design of portable devices. I am happy to recommend publication of the review article after revision.

Author Response

Dear Reviewer,

We very much appreciated the encouraging, critical and constructive comments on this manuscript by the reviewer. The comments have been useful in improving the manuscript. We have revised our manuscript according to your comments. Below we give point by point answers. The changes in the manuscript were highlighted in yellow.

  1. The manuscript has numerous grammatical and linguistic errors and should be checked and revised thoroughly. There are parts that the message is not only unclear but also highly confusing. Eg. Abstract: “…the place of catalytic hydrolysis of NaBH4 with the using…”; “…the prospect of using alkali-free solid-phase compositions…”; etc. This is not an exhaustive list since similar errors occur throughout the manuscript.

According your recommendation, the text was edited for errors.

  1. Page 2, line 88: “…examples of irreversible “off-board”…”. This part is confusing for the readers. What exactly ”off-board” means in this case? Please revise or provide additional explanation.

Thank you so much for your suggestion. The “off board” and “on board” terms are widely used in discussed research area. To meet this comment the sentence “Table 1 lists the main examples of irreversible “off-board” hydrogen storage based on hydrolysis or involvement of water in the hydrogen generation process.” was changed to “Table 1 lists the main hydrogen storage systems based on hydrolysis or involvement of water in the hydrogen generation process, in which the spent fuels or byproducts should be removed from the reactor and “off board” regenerated.” (Lines 93-95)

  1. Conclusions, line 729: “…heat and mass transfer and foaming on the kinetics of hydrogen…” This is slightly confusing. The foaming I would assume is due to exothermicity of the reaction and the parameters that might have some correlation with the kinetics of the H2 production rather than the actual foaming of the system. I: would suggest the authors revising this part.

Thank you very much for your comment. This part was revised.

“Dosing a limited amount of water provides high GHSC values as compared to the hydrolysis of NaBH4 alkaline solutions in flow reactor. But in this case there are problems of self-heating and foaming the reaction medium caused by exothermicity of the reaction and vigorous gas formation. For estimation of the potential of such hydrogen generation process, it is necessary to perform further experimental and theoretical studies on the effect of heat and mass transfer on the kinetics of hydrogen generation and its purity when optimizing the design of such generators and varying the conditions of their operating.” (Lines 742-749)

Reviewer 2 Report

Ms No: Catalysts – 1109911

Title: Recent advances in applications of Co-B catalysts in NaBH4-based portable hydrogen generators              

Authors: Valentina I. Simagina, Anna M. Ozerova, Oksana V. Komova and Olga V. Netskina

Evaluation

The present paper deals with the recent advances in Co-B catalysts in NaBH4 – based portable hydrogen generators. The present work is well written and the authors are making a great effort in order to elucidate specific aspects of the aforementioned issue. It is true that there is a lack of publication as it concerns the Co catalysts and the present work aims to fill this gap. Based on the comments listed below, my opinion is that the paper is suitable for publication after minor revisions.

Comments

  1. At the first sections authors are referring to various methods for hydrogen production from NaBH4. Among them the highest yield is achieved by the noble metal catalysts (section 4). In the entire paper there is a lack of yield numbers as it concerns the H2 production. Authors should add some representative values in order to help the reader to understand the variations in the hydrogen production yield among the different applied methods.
  2. In section 4 in some of the works that the authors are referring to, is there any further characterization (i.e. XPS, FTIR) related to the initial state of the noble metal used. It would be extremely interesting to be mentioned since it is directly related to the reaction mechanism as well as the hydrogen yield.
  3. If it is possible authors should add more information (if applicable) related to the role of the precursors used for the Co based catalytic materials in order to elucidate that particular aspect. If there isn’t any information what is the authors opinion about this.
  4. Does the reactor material play any significant role in the entire performance?
  5. As it is well known the PEMFC are high performance devices. The combination with NaBH4 – based hydrogen generators seems to be quite promising. Is there any work in the literature presenting a techno-economical analysis for the whole process? If there is, authors should mention it and add a short text. I understand that this is not the issue of the present review paper but yet is quite interesting and since the analysis is related to a subject relative to hydrogen production the economical status of the proposed process should at some point be mentioned.
  6. In Table 1 authors are referring to the MgH2 material (ref. [54]) and in the advantages they are mentioning that is non-toxic. In fact it is the only material in the table that is non-toxic. To the best of our knowledge are there any other materials used for the NaBH4 that are non-toxic?
  7. Authors should add colors in the schematic diagram demonstrated in Fig. 3 as in the case of the schematic diagram displayed in Fig. 4 in order to aesthetically improve the result.
  8. In Figure 2 authors should clarify what we actually see. I understand that this figure is adapted but authors should ease the reader to comprehend the significance of the COCl2 average particle size and the correlation with the H2 yield.

Author Response

Answer to Reviewer2

Dear Reviewer,

We very much appreciated the encouraging, critical and constructive comments on this manuscript by the reviewer. The comments have been thorough and useful in improving the manuscript. We have revised our manuscript according to your comments. Below we give point by point answers to your comments and questions. The changes in the manuscript were highlighted in yellow.

  1. At the first sections authors are referring to various methods for hydrogen production from NaBH4. Among them the highest yield is achieved by the noble metal catalysts (section 4). In the entire paper there is a lack of yield numbers as it concerns the H2 Authors should add some representative values in order to help the reader to understand the variations in the hydrogen production yield among the different applied methods.

Activity of the different catalysts is difficult to compare because it depends on different parameters such as temperature, concentration of NaBH4 and NaOH, amount of catalysts in the reaction medium, and heat and mass transfer in the reactor. Expression of these values as mlH2×min-1×g-1cat doesn't always give correct information if the reaction conditions differ. But, it is necessary to make these estimations in order to compare the catalysts. Such comparison of activity of noble and non-noble catalysts was discussed in [81, 82]. It can be seen that the activity of non-noble Co-B-based catalysts (15,000-26,000 mlH2×min-1×g-1cat) reaches the activity of Ru or Pt catalysts. The researches on Rh catalysts are restricted by their high cost.

According to Reviewer's comments the following text was added to Section 4
(Line 222-223, 226-229).

“The researches on Rh catalysts are restricted by their high cost. … The comparison of activity of noble and non-noble catalysts was discussed in [81, 82]. It can be seen that the activity of non-noble Co-B-based catalysts (15,000-26,000 mlH2×min-1×g-1cat) reaches the activity of Ru or Pt catalysts.”

  1. In section 4 in some of the works that the authors are referring to, is there any further characterization (i.e. XPS, FTIR) related to the initial state of the noble metal used. It would be extremely interesting to be mentioned since it is directly related to the reaction mechanism as well as the hydrogen yield.

To meet this comment the following text was added to Section 4 (Lines 223-226).

“Investigation of the catalytically active phase formed from compounds of ruthenium, rhodium, platinum, and palladium in NaBH4 medium has shown that it consists of agglomerated nanosized particles with a metal crystal lattice [98-100]. Residual amount of boron was presented in Ru metal [101].”

  1. If it is possible authors should add more information (if applicable) related to the role of the precursors used for the Co based catalytic materials in order to elucidate that particular aspect. If there isn’t any information what is the authors opinion about this.

Thank you for this question. This topic is partially discussed in this review. In Section 5.1, the rate of reduction of cobalt salts and cobalt oxide in the reaction medium of NaBH4 with the formation of catalytically active Co-B phase is compared. Also, in Section 5.4 reactivity of cobalt borate and cobalt hydroxide under action of NaBH4 is discussed. The references on the influence of cobalt salt precursor were cited in Section 5.1. They are [91,114,119,122,125] (Line 269). Analysis of this literature didn't give unambiguous answer to this question. Further researches are needed to perform to elucidate this issue.

  1. Does the reactor material play any significant role in the entire performance?

This wasn't the issue of the present review paper.

But, you're right. To increase competitiveness of hydrogen economy it necessary to develop compact energy devices to meet the requirements of light weight, small size, and high power density. On the other hand, to avoid self-heating of reaction medium and to control hydrogen generation rate materials with the high heat transfer properties should be applied. Also, the reactor material should capable of withstanding pressure and hydrogen embrittlement. So, the optimization of reactor material is one of the key tasks. The articles analyzed in Section 7 provide the data for reactors mainly made of heat-conducting steel. Their masses weren't indicated by the authors, and these data are difficult to compare.

To meet this comment the following text was added to Section 7 (Lines 620-625).

“Note that the optimization of reactor material of hydrogen generator is one of the key tasks. On the one hand, the reactor material should meet the requirements of light weight, small size, and high power density of energy device. On the other hand, to avoid self-heating of reaction medium and to control hydrogen generation rate materials with high heat transfer properties should be applied. Also, the reactor material should capable of withstanding pressure and hydrogen embrittlement.”

  1. As it is well known the PEMFC are high performance devices. The combination with NaBH4– based hydrogen generators seems to be quite promising. Is there any work in the literature presenting a techno-economical analysis for the whole process? If there is, authors should mention it and add a short text. I understand that this is not the issue of the present review paper but yet is quite interesting and since the analysis is related to a subject relative to hydrogen production the economical status of the proposed process should at some point be mentioned.

This is a rather complex issue that requires a comprehensive consideration of the proposed hydrogen storage and generation technology and its comparison with others.

NaBH4 hydrolysis is extensively studied owing to its advantages for hydrogen storage but it exhibits some drawbacks such as a high costs of NaBH4 and high energy consumption at regeneration process of the by-products (sodium borates). These parameters depend not only on efficiency of the applied industrial technologies of chemicals production but also on cost of raw material, electricity, annual production of NaBH4, and etc.

According to this, the citing the manuscript on techno-economical analysis of this process was added in the Introduction. (Lines 59-65)

“A model where hydrogen is produced through the electrolysis of water, taking advantage of the electrical energy produced by a renewable generator (photovoltaic panels) and chemically stored by the synthesis of NaBH4 is discussed in [19]. It was shown that due to safety and high volumetric density of NaBH4 it will be promising technology to transportation to remote sites where hydrogen is released from NaBH4 hydrolysis and used for energy production.”

  1. In Table 1 authors are referring to the MgH2 material (ref. [54]) and in the advantages they are mentioning that is non-toxic. In fact it is the only material in the table that is non-toxic. To the best of our knowledge are there any other materials used for the NaBH4 that are non-toxic?

Thanks the Reviewer for this comment. Indeed, all water reactive materials presented in Table 1, as well as MgH2, are toxic to humans. They are harmful if swallowed and inhaled, and may cause skin and eye irritant.

To meet this comment the MgН2 advantage of being non-toxic was removed from the Table 1.

  1. Authors should add colors in the schematic diagram demonstrated in Fig. 3 as in the case of the schematic diagram displayed in Fig. 4 in order to aesthetically improve the result.

That was done.

  1. In Figure 2 authors should clarify what we actually see. I understand that this figure is adapted but authors should ease the reader to comprehend the significance of the CоCl2 average particle size and the correlation with the H2

The capture of Figure 2 was revised. Phrase “Co-B catalyst instantaneously formed in the NaBH4 hydrolysis medium has a smaller particle size and shows a higher hydrogen generation rate.” was added.

Also, the phrase “A lower rate of cobalt chloride reduction in NH3BH3 medium is confirmed by the presence of an induction period on the hydrogen generation curve (Figure 2).” was added in Section 5.1. (Lines 257-259)
